# Oligomer Formation by Physiologically Relevant C-Terminal Isoforms of Amyloid *β*-Protein

**DOI:** 10.3390/biom14070774

**Published:** 2024-06-28

**Authors:** Rachit Pandey, Brigita Urbanc

**Affiliations:** Department of Physics, Drexel University, Philadelphia, PA 19104, USA; rp878@drexel.edu

**Keywords:** amyloid β-protein, Alzheimer’s disease, discrete molecular dynamics

## Abstract

Alzheimer’s disease (AD) is a neurological disorder associated with amyloid β-protein (Aβ) assembly into toxic oligomers. In addition to the two predominant alloforms, A$\beta_{1-40}$ and A$\beta_{1-42}$, other C-terminally truncated Aβ peptides, including A$\beta_{1-38}$ and A$\beta_{1-43}$, are produced in the brain. Here, we use discrete molecular dynamics (DMD) and a four-bead protein model with amino acid-specific hydropathic interactions, DMD4B-HYDRA, to examine oligomer formation of A$\beta_{1-38}$, A$\beta_{1-40}$, A$\beta_{1-42}$, and A$\beta_{1-43}$. Self-assembly of 32 unstructured monomer peptides into oligomers is examined using 32 replica DMD trajectories for each of the four peptides. In a quasi-steady state, A$\beta_{1-38}$ and A$\beta_{1-40}$ adopt similar unimodal oligomer size distributions with a maximum at trimers, whereas A$\beta_{1-42}$ and A$\beta_{1-43}$ oligomer size distributions are multimodal with the dominant maximum at trimers or tetramers, and additional maxima at hexamers and unidecamers (for A$\beta_{1-42}$) or octamers and pentadecamers (for A$\beta_{1-43}$). The free energy landscapes reveal isoform- and oligomer-order specific structural and morphological features of oligomer ensembles. Our results show that oligomers of each of the four isoforms have unique features, with A$\beta_{1-42}$ alone resulting in oligomers with disordered and solvent-exposed N-termini. Our findings help unravel the structure–function paradigm governing oligomers formed by various Aβ isoforms.

## 1. Introduction

Alzheimer’s disease (AD) is a severe neurological disorder associated with a progressive memory loss, buildup of extracellular amyloid plaques, and intracellular neurofibrillary tangles, accompanied by a significant loss of neurons in the brain. Despite significant research findings that span several decades, there is currently no known cure for AD. The most widely accepted hypothesis on the cause of this disease is the amyloid cascade hypothesis, developed in 1992 Hardy and Higgins [1] and revised in 2002 [2], which in its revised form postulates that AD is triggered by the formation of toxic, predominantly unstructured assemblies, oligomers, formed by amyloid β-protein (Aβ) [3,4].

The vast majority of Aβ in the brain is produced from a transmembrane amyloid precursor protein (APP) through sequential cleavages by β- and γ-secretases [5,6,7]. β-secretase cleaves the APP in the extracellular region, producing the C-terminal fragment of APP with its N-terminal region corresponding to the Aβ sequence. Follow-up γ-secretase actions on the resulting C-terminal fragment of APP give rise to several C-terminally distinct Aβ isoforms, including the predominant A$\beta_{1-40}$ and A$\beta_{1-42}$ [8]. Although the human body produces more A$\beta_{1-40}$ than A$\beta_{1-42}$, A$\beta_{1-42}$ (but not A$\beta_{1-40}$) levels as well as the A$\beta_{1-42}$/A$\beta_{1-40}$ ratio in cerebrospinal fluid (CSF) are used as biomarkers for AD [9]. A$\beta_{1-42}$ is generally considered to be the most toxic of the Aβ isoforms. Several studies showed that in addition to A$\beta_{1-40}$ and A$\beta_{1-42}$, other Aβ isoforms may play a role in AD [10,11,12,13,14,15]. Saito and collaborators reported early memory impairment in a mouse model of AD, which was genetically modified to produce human A$\beta_{1-43}$ [11]. A$\beta_{1-43}$ has been reported to aggregate faster and be more neurotoxic than A$\beta_{1-42}$ [10,11]. Lauridsen et al. showed that A$\beta_{1-43}$ as a biomarker in CSF distinguishes between mild cognitive disorder and AD better than A$\beta_{1-42}$ [12]. A more recent study examining Aβ isoforms as biomarkers for AD elucidated the importance of A$\beta_{1-37}$, showing that the A$\beta_{1-42}$/A$\beta_{1-37}$ ratio outperformed the A$\beta_{1-42}$/A$\beta_{1-40}$ ratio [13]. A similar study highlighted a potentially important role of A$\beta_{1-38}$ as an AD biomarker in CSF [15]. An in vitro study by Braun and collaborators demonstrated that A$\beta_{1-37}$, A$\beta_{1-38}$, and A$\beta_{1-40}$ individually inhibit A$\beta_{1-42}$ aggregation, whereby cooperatively, a mixture of the three shorter Aβ isoforms more significantly impeded A$\beta_{1-42}$ aggregation than each individual isoform [14]. Another in vitro study revealed distinct biochemical characteristics of A$\beta_{1-40}$, A$\beta_{1-42}$, and A$\beta_{1-43}$ when interacting with apolipoprotein E (apoE) [16].

We here posit that oligomer formation is sensitive to C-terminal truncation and that A$\beta_{1-38}$, A$\beta_{1-40}$, A$\beta_{1-42}$, and A$\beta_{1-43}$ are characterized by isoform-specific oligomer size distributions and conformational ensembles. To test this hypothesis, we examine oligomer formation by all four Aβ isoforms by discrete molecular dynamics (DMD) simulations. Event-driven DMD is a powerful tool for studying peptide and protein self-assembly [17] and can be combined with various implicit-solvent peptide models using different levels of coarse graining, from a minimal tetrahedron model representing the entire protein [18] through one-bead-per-residue [19,20], four-beads-per-residue [21,22,23,24], and all-atom models [25,26,27,28].

We here examine A$\beta_{1-38}$, A$\beta_{1-40}$, A$\beta_{1-42}$, and A$\beta_{1-43}$ oligomer formation by DMD combined with an intermediate-resolution protein model with four beads per residue and amino acid residue-specific interactions (DMD4B-HYDRA force field), which was applied previously to characterize isoform-specific oligomer formation and conformational ensembles formed by A$\beta_{1-40}$ and A$\beta_{1-42}$ [23], their Arctic mutants [29], N-terminally truncated isoforms [30], and [K16A] and [K28A] mutants [31]. DMD4B-HYDRA simulations revealed A$\beta_{1-40}$ oligomer size distribution dominated by dimers with decreasing levels of trimers and tetramers. In contrast, A$\beta_{1-42}$ oligomer size distribution was multimodal with a global maximum at dimers to trimers, a second maximum at pentamer and hexamers, and a third (the least prominent) maximum at decamers through dodecamers [23,29], consistent with in vitro study [32,33]. The process of Aβ oligomer formation in these studies traces self-assembly of 32 unstructured monomeric peptides, which simultaneously fold and self-assemble into a state with a quasi-steady-state oligomer size distribution. To feed the statistical analysis and derive a reliable oligomer size distribution, the number of replica DMD4B-HYDRA trajectories was set to eight. In this study, we expanded the number of trajectories to 32 to obtain a better statistical sampling of different stages of A$\beta_{1-38}$, A$\beta_{1-40}$, A$\beta_{1-42}$, and A$\beta_{1-43}$ oligomer formation to facilitate the comparison among these four physiologically relevant Aβ isoforms. Our study reveals that oligomer formation propensity increases with the number of amino acid residues in the C-terminal region and elucidates distinct isoform-specific structural characteristics of oligomer conformational ensembles.

## 2. Materials and Methods

### 2.1. DMD4B-HYDRA Simulations

#### 2.1.1. DMD

DMD is a physics-based molecular dynamics method using discontinuous square well-type potentials, which results in event-driven dynamics [34]. By employing a combination of spherically symmetric square-well potentials instead of continuous potentials between pairs of particles, alongside a coarse-grain protein model with an implicit-solvent force field, it is possible to substantially increase the number of peptides in each trajectory and examine protein self-assembly on timescales that are significantly larger than the typical timescales used in all-atom explicit-solvent MD [17,35]. The successful adaptation of this method has enabled protein modeling [19,36,37,38] and facilitated studies on the folding and aggregation of various proteins, including a three-helix-bundle protein [39,40], the SH3 protein [19,38,41], and Aβ [22].

#### 2.1.2. Four-Bead Protein Model


In our simulations, we employ a four-bead protein model initially developed by Ding et al. [21]. This model replaces each amino acid with up to four beads: the amide N, the α-carbon ($C_{\alpha }$), the carbonyl $C^{\prime }$ (CO) group, and the side-chain β-carbon ($C_{\beta }$) group. The four-bead model accounts for backbone hydrogen bonding, which involves interactions between the carbonyl oxygens and amide hydrogens of different amino acid residues. Due to the absence of explicit hydrogens and oxygens in the four-bead protein model, their positions can be inferred based on the neighboring N- and $C^{\prime }$-atoms. The hydrogen bond is then modelled as a multiatom interaction that involves the distance of the N-atom of one residue and the $C^{\prime }$-atom of another residue as well as the two neighboring beads on the N-atom and $C^{\prime }$-atom in order to model the directionality of the hydrogen bond.

#### 2.1.3. Residue-Specific Interactions Due to Hydropathy


To avoid the modelling of an explicit solvent and thereby increase the timescale of DMD simulations, the hydrophobic attraction and hydrophilic repulsion between pairs of $C_{\beta }$ beads are incorporated into the force field to account for the hydrophobic and hydrophilic effects that water imparts on proteins. The effective short-range hydrophobic attraction and hydrophilic repulsion are implemented through a simple square-well potential with a cutoff distance of 0.75 nm, as described previously [17,23]. The amino acid residue hydropathy scale used in DMD4B-HYDRA is due to Kyte and Doolittle [42], whereby residues of Ile, Phe, Cys, Val, Ala, and Met are treated as hydrophobic, whereas His, Gln, and Asn are modelled as uncharged hydrophilic and Arg, Lys, Asp, and Glu are treated as charged hydrophilic. The remaining amino acids with low hydropathy values are not subjected to this residue-specific hydropathic potential. Whereas effective short-range electrostatic interactions are implemented in this DMD4B-HYDRA force field and have been used in some previous applications to model folding of two nonglycosylated mucin domains [43] and oligomer formation by an amyloidogenic isoform of stefin-B [24], previous applications of DMD4B-HYDRA to oligomer formation of Aβ peptides [29,44] demonstrated that that incorporating electrostatic interactions into DMD simulations led to the formation of large oligomers, inconsistent with experimental findings [32,33].

#### 2.1.4. Units in DMD4B-HYDRA Force Field


The energy associated with the backbone hydrogen bond interaction, denoted as $E_{HB}$, serves as an energy unit in DMD4B-HYDRA simulations. The temperature in our simulations is expressed in units of $E_{HB}/k_{B}$. Our simulations are performed at T=0.13 as an estimate of physiological temperature 310 K; thus, $E_{HB}\approx 4.6$ kcal/mol [29,43,45]. By using the equipartition theorem, $\frac{1}{2}mv^{2}=\frac{1}{2}k_{B}T$, we can calculate the time unit of DMD4B-HYDRA simulations by taking into account the spatial resolution of 0.1 Å. In the equation above, *m* represents the unit mass used in the DMD simulations, which is equal to one-quarter of the mass of alanine. By substituting the estimated value $m\approx 3\times 10^{-26}$ kg into the equation, we can calculate a 1D root mean square velocity of the beads, 300 m/s, resulting in Δt≈0.03 ps. In our study, the total time per trajectory is $40\times 10^{6}$ time units, which corresponds to 1.2 μs. The strength of the hydropathic interactions is set to $E_{HP}=0.3E_{HB}\approx 1.4$ kcal/mol, which corresponds to the effective attractive interaction between two Ile side chains, as Ile is the most hydrophobic residue in the Kyte–Doolittle hydropathy scale.

#### 2.1.5. Simulation Protocol


All DMD4B-HYDRA simulations were performed at a constant volume and temperature using a Berendsen thermostat [46]. For each of the four isoforms, i.e., A$\beta_{1-38}$, A$\beta_{1-40}$, A$\beta_{1-42}$, and A$\beta_{1-43}$, 32 peptides were initially placed into a cubic lattice in a cubic simulation box of $25^{3}$
${\mathrm{nm}}^{3}$. To obtain 32 independent starting conformations for each of the four isoforms, we acquired a short simulation at a high temperature T=4, during which the hydrogen bonding and effective residue-specific interactions were turned off to prevent Aβ folding and self-assembly. For each of the four isoforms, the 32 independent starting conformations contained 32 unstructured spatially separated Aβ monomers. The production runs for each isoform consisted of 32 distinct replica trajectories acquired at physiological temperature, T=0.13 $E_{HB}$/$k_{B}$. For each of the four isoforms, we collected 32 40 $\times \;10^{6}$ simulation time unit-long trajectories, resulting in total in 4×32×1.2μs =153.6μs =0.1536 ms of DMD4B-HYDRA simulations.

### 2.2. Structural Analysis


#### 2.2.1. Oligomer Size Distributions


Oligomers are identified based on the threshold distance between pairs of peptides. When the distance between any two beads that belong to two different peptides is less than 5 Å, the two peptides belong to the same oligomer. At a specified simulation time, we calculate for each of the 32 trajectories both molar and mass-weighted oligomer size distributions from the arrangement of 32 Aβ peptides in the simulation box. Such an arrangement consists of 32 peptides that are distributed in varying numbers of clusters of various sizes, ranging from monomers, dimers, … up to 32-mers. To calculate a molar size distribution at a specific simulation time, we count the number of oligomers of each size and divide this number by the total number of oligomers in the simulation box for each trajectory. To obtain the corresponding mass-weighted oligomer size distribution, the number of oligomers of size *n* is multiplied by *n*, and the entire distribution is divided by the total number of peptides in the simulation box (i.e., 32) to obtain a probability that a peptide is within an oligomer of size *n*. The final molar and mass-weighted propensity are calculated as ensemble averages over all 32 trajectories. The error bars correspond to the standard error of the mean (SEM) values.

#### 2.2.2. Secondary Structure Distributions


The secondary structure was analyzed by using the STRIDE algorithm [47] integrated within the Visual Molecular Dynamics [48] (VMD) software package. The per-residue turn, strand, and coil propensities were first calculated as averages over 32 peptides (regardless of their assembly state), and then time–averaged within each trajectory using 200 time frames between $20\times 10^{6}$ and $40\times 10^{6}$ time units, separated by $10^{5}$ time units. The averaged per-residue secondary structure propensities and the respective standard error of the mean (SEM) values were then calculated from 32 resulting per-residue propensities by an ensemble average over 32 independent trajectories. Similar calculation was performed for each assembly state of each isoform. When calculating per-residue turn, strand, and coil propensities for each assembly state, all monomers through hexamers within $20\times 10^{6}$–$40\times 10^{6}$ simulation time units of each trajectory were first used to derive the average propensities followed by the ensemble average over 32 trajectories, which also allowed us to calculate the SEM values, reflecting trajectory-to-trajectory variability. We also calculated the average strand, turn, and coil content per peptide in Aβ monomers and oligomers up to and including heptamers. Using all 32 trajectories, ensembles of 32 peptides at time frames between $20\times 10^{6}$ and $40\times 10^{6}$ time units, were separated into monomers and oligomers up to and including heptamers. For each assembly state, the average strand, turn, and coil content per peptide and the respective SEM values were calculated by averaging over all peptides in a conformation and over all conformations.

#### 2.2.3. Contact Maps


To gather the information regarding the tertiary and quaternary structure, we calculated contact maps for both intra- and intermolecular interactions. A precise criterion was established to define the interaction between amino acids—a pairwise distance of 0.5 nm or less between their centers of mass. This criterion was then used to construct a contact map, a matrix presenting a comprehensive view of amino acid interactions. Each element (i,j) of this matrix represents the average count of interactions between amino acid i and amino acid j. A crucial aspect of our analysis was the categorization into intra- and intermolecular contact maps. This classification was based on whether amino acids i and j were part of the same peptide chain or not. In cases where i and j were constituents of a single peptide, their interactions were classified as belonging to the intramolecular contact map. On the other hand, if these amino acids spanned across distinct peptides, their interactions contributed to the intermolecular contact map.

#### 2.2.4. Solvent-Accessible Surface Area (SASA) and N-C Distance


The hydrophobic SASA is the total area of hydrophobic region exposed to the solvent. It was calculated for a 20 $\times \;10^{6}$–40 $\times \;10^{6}$ time frame of all 32 trajectories for the oligomers of all four variants, A$\beta_{1-38}$, A$\beta_{1-40}$, A$\beta_{1-42}$, and A$\beta_{1-43}$, using VMD software. To accomplish this, a spherical surface was constructed around each atom, positioned 1.4 Å away from the atom’s Van der Waals surface. By combining these individual spherical surfaces, a comprehensive 3D surface representation was obtained for each amino acid. The SASA was subsequently determined by calculating the portion of this surface area that did not overlap with the surfaces of neighboring amino acids. This nonoverlapping region represents the accessible surface area available to the solvent.

The N-C distance is the length of the vector that connects the N-terminal $C_{\alpha }$ atom and C-terminal $C_{\alpha }$ atom. The distance from the N-terminal D1 to C terminal T43 (A$\beta_{1-43}$) or A42 (A$\beta_{1-42}$) or V40 (A$\beta_{1-40}$) or G38 (A$\beta_{1-38}$) for each oligomer size of each of the four variants was calculated.

#### 2.2.5. Potential of Mean Force (PMF)


The PMF, which represents the free energy landscape, was computed using two reaction coordinates: hydrophobic SASA ($X_{1}$) and the N-C distance ($X_{2}$). The reaction coordinates for each variant was calculated and then averaged across all the peptides within an oligomer to obtain the PMF per oligomer of that variant. The free energy was calculated as W = $W_{0}$− ln P($X_{1}$,$X_{2}$), where P($X_{1}$, $X_{2}$) is the normalized distribution of the two reaction coordinates. $W_{0}$ is defined as −ln(N + 1), with N representing the total number of conformations. To obtain P($X_{1}$, $X_{2}$), a 2D histogram of the two reaction coordinates was generated, and it was then normalized by the total number of peptides used in the reaction coordinate calculations. The lowest values in the PMF correspond to the most probable conformations.

#### 2.2.6. Principal Moments of Inertia


The three-dimensional shapes of oligomers can be effectively characterized by calculating their principal moments of inertia. In our study, we used the implementation available in VMD and applied it to monomer to decamer conformations of each of the four Aβ isoforms formed within 20 $\times \;10^{6}$ to 40 $\times \;10^{6}$ simulation time units. Through the computation of the eigenvalues of the inertia tensor, we derived three principal moments of inertia, $I_{1}$, $I_{2}$, and $I_{3}$, for each oligomer size of each Aβ isoform. The moments of inertia $I_{1}$, $I_{2}$, and $I_{3}$ are normalized with the mass of a monomer, $M_{0}$. If n is the order of a oligomer and M is the total mass of the oligomer, then M = n.$M_{0}$.

## 3. Results

Many previous studies have shown that the difference at the C-terminus between A$\beta_{1-40}$ and A$\beta_{1-42}$ results in distinct conformational ensembles and distinct oligomerization pathways [23,29,49,50,51], whereas oligomer formation of A$\beta_{1-38}$ and A$\beta_{1-43}$ has not been examined to date. We here posit that Aβ oligomer formation and structure is highly sensitive to the C-terminal sequence of Aβ. To test this assertion, we examine oligomer formation pathways and structurally characterize conformational ensembles of oligomers formed by four physiologically relevant isoforms that differ at the C-terminus: A$\beta_{1-38}$, A$\beta_{1-40}$, A$\beta_{1-42}$, and A$\beta_{1-43}$. The primary structure of the 43-residues-long A$\beta_{1-43}$ is $D_{1}$${\mathrm{AEFRHDSGYE}}_{11}$${\mathrm{VHHQKLVFFA}}_{21}$${\mathrm{EDVGSNKGAI}}_{31}$${\mathrm{IGLMVGGVVI}}_{41}$AT, and A$\beta_{1-38}$, A$\beta_{1-40}$, and A$\beta_{1-42}$ sequences lack the five, three, and one C-terminal residues of A$\beta_{1-43}$, respectively. In the following, the peptide regions $L_{17}$${\mathrm{VFFA}}_{21}$ and $I_{31}$${\mathrm{IIGLMV}}_{36}$ are referred to as the central hydrophobic cluster (CHC) and midhyrophobic region (MHR), respectively, whereas the region that separates the CHC and MHR, $A_{21}$${\mathrm{EDVGSNKGA}}_{30}$, is referred to as the central folding region (CFR). The peptide region $V_{39}$$V_{40}$$I_{41}$$A_{42}$$T_{43}$ (or the corresponding C-terminally truncated version), which is absent from ${A\beta }_{1-38}$ and fully present only in A$\beta_{1-43}$, is referred to as the C-terminal region (CTR).

For each of the four Aβ isoforms, we perform DMD4B-HYDRA simulations, whereby each system contains 32 initially unstructured Aβ monomers (see Section 2.1.5 in Section 2). The protocol employed in this study differs from the previously published protocol, in which 8 replica trajectories per peptide were used [23,29,30,31]. Here, we acquired in total 32 trajectories for each of A$\beta_{1-38}$, A$\beta_{1-40}$, A$\beta_{1-42}$, and A$\beta_{1-43}$, resulting in 128 trajectories, to improve the statistics of the structural analysis.

### 3.1. Oligomer Formation Propensity Increases with the Number of Residues at the C-Terminus

Previous DMD4B-HYDRA simulations have shown that A$\beta_{1-40}$ and A$\beta_{1-42}$ self-assemble into oligomers of different sizes and that the oligomer size distributions reach a quasi-steady state, during which these distributions remain intact or change over time only minimally [29]. It should be noted that there are two ways to quantify the oligomer size distributions, by deriving a molar distribution or mass-weighted distribution (see Section 2 for details). Previous DMD4B-HYDRA studies reported molar oligomer size distributions of various Aβ isoforms, in which the number of oligomers of a specific size is normalized by the total number of oligomers in the ensemble [23,29,30,31]. The importance of the oligomer size distributions derived from simulations stems from the availability of experimental data on various Aβ peptides [32,33] to which the computational results can be compared. However, it remains uncertain which of the two types, molar or mass-weighted distribution (the latter gives the probability of finding a peptide in an oligomer of a particular size), can provide a more realistic comparison to in vitro oligomer size distribution obtained through chemical cross-linking combined with gel electrophoresis (SDS-PAGE), followed by silver staining of the gel bands, whereby the resulting intensities of silver staining are used to experimentally obtain the oligomer size distributions [32,33,52,53]. Both types of distribution are evidently interrelated, whereby the advantage of using the mass-weighed distribution is in its bias toward the larger oligomers.

We calculated both the molar and mass-weighted oligomer size distributions of each of the four isoforms and monitored their time evolution. Figure 1 shows the time evolution of A$\beta_{1-38}$, A$\beta_{1-40}$, A$\beta_{1-42}$, and A$\beta_{1-43}$ mass-weighted oligomer size distributions for all four isoforms at different stages of assembly, corresponding to simulation time units of 0, $10\times 10^{6}$, $20\times 10^{6}$, $30\times 10^{6}$, and $40\times 10^{6}$. For comparison, Appendix A shows the time evolution of the respective molar oligomer size distributions.

Results in Figure 1 indicate that during the first $10\times 10^{6}$ simulation time units, the fast hydrophobic collapse of initially monomeric systems results in the most significant changes in the mass-weighted oligomer size distributions of the four Aβ isoforms. After $20\times 10^{6}$ simulation time units, the changes in the oligomer size distributions steadily subside, indicating that oligomer formation of all four isoforms approaches a quasi-steady state, during which the changes in the oligomer size distributions are significantly slower than during the initial hydrophobic collapse. Similar observations can be made on the corresponding molar oligomer size distributions displayed in Appendix A, demonstrating that systems are in a quasi-steady state between $20\times 10^{6}$ and $40\times 10^{6}$ simulation time units.

All four mass-weighted and molar oligomer size distributions derived from trajectories at $40\times 10^{6}$ simulation time units are shown in Figure 2 and Appendix A, respectively, to facilitate an easier comparison among the four isoforms. It is clear that A$\beta_{1-38}$ and A$\beta_{1-40}$ form very similar distributions that are quite distinct from A$\beta_{1-42}$ and A$\beta_{1-43}$ distributions. A$\beta_{1-42}$ and A$\beta_{1-43}$ distributions are also quite similar to each other. Nonetheless, these distributions are isoform-specific as well. The mass-weighted oligomer size distribution of A$\beta_{1-38}$ is dominated by dimers and trimers alongside an increased abundance of septamers. The mass-weighted A$\beta_{1-40}$ oligomer size distribution is unimodal and dominated by trimers. As shown in Appendix A, the corresponding molar A$\beta_{1-38}$ and A$\beta_{1-40}$ oligomer size distributions are similar to the respective mass-weighted distributions but are dominated by dimers (in the case of A$\beta_{1-38}$) and dimers and trimers (in the case of A$\beta_{1-40}$). The mass-weighted distributions of two longer peptides are both multimodal. Mass-weighted A$\beta_{1-42}$ distribution has a global maximum at trimers to tetramers and additional maxima at hexamers, unidecamers, and pentadecamers, whereas the corresponding A$\beta_{1-43}$ distribution has a global maximum at trimers and increased hexamer and octamer propensities alongside another maximum between penta- and hexadecamers (Figure 2). The molar A$\beta_{1-42}$ distribution retains a clear multimodal character with three maxima at trimers, hexamers, and unidecamers (Appendix A). In contrast, the molar A$\beta_{1-43}$ distribution is dominated by trimers and an increased octamer propensity but without additional peaks at larger oligomers (Appendix A).

Notably, despite the improved statistics with 32 (instead of 8) independent trajectories used in the present work, the molar A$\beta_{1-40}$ and A$\beta_{1-42}$ oligomers size distributions are consistent with and thus validate previously published oligomer size distributions, i.e., a unimodal A$\beta_{1-40}$ distribution with a single maximum at dimers and a multimodal molar A$\beta_{1-42}$ distribution with maxima at dimers, hexamers, and decamers to dodecamers [23,29]. The results of the present analysis indicate that the two shorter peptides form, on average, significantly smaller oligomers than the two longer peptides. Oligomer formation propensity appears to correlate with the number of amino acid residues at the C-terminus of Aβ, whereas the oligomer size distributions exhibit isoform-specific characteristics.

### 3.2. Selection of Conformations for Structural Analysis


The analysis of the oligomer size distributions reveals that the oligomer size distributions do not significantly change with time between $20\times 10^{6}$ and $40\times 10^{6}$ simulation time units. Appendix A shows the total number of oligomers of different sizes, ranging from monomers to decamers, produced by DMD4B-HYDRA simulations in the time range from $20\times 10^{6}$ to $40\times 10^{6}$ simulation time units, during which the oligomer size distributions are quasi-stable. These conformations were structurally analyzed, and the results of these analyses are described below.

### 3.3. Aβ Isoform-Specific Secondary Structure Characteristics


We here analyze the average secondary structure per amino acid residue in A$\beta_{1-38}$, A$\beta_{1-40}$, A$\beta_{1-42}$, and A$\beta_{1-43}$ conformational ensembles collected over 32 trajectories between 20 $\times \;10^{6}$ and 40 $\times \;10^{6}$ simulation time steps, regardless of the assembly states, as described in Section 2. Figure 3 shows per-residue propensities for the three predominant secondary structure elements: turn, strand, and coil. The error bars in Figure 3 are very small, indicating that the average per-residue secondary structure propensities, obtained by averaging over 32 peptides in each time frame and over 200 time frames, do not show much trajectory-to-trajectory variability. We also calculated per-residue turn, strand, and coil propensities for each assembly state separately (Appendix A) to show that dimers through hexamers of each isoform exhibit very similar secondary structure propensities, which elucidates oligomers as self-similar hydrophobically collapsed assemblies. Nonetheless, these propensities exhibit isoform-specific features described below.

Results in Figure 3A shows that turn propensities of the four isoforms are very similar within in the H13-M35 range, whereas Aβ isoform-specific deviations are notable at the two termini. Overall, there are six turn regions observed in A$\beta_{1-40}$, A$\beta_{1-42}$, and A$\beta_{1-43}$ conformational ensembles. At the C-terminus, where differences among the four isoforms would be expected, A$\beta_{1-38}$ lacks the turn region at V36-V39. Less expected are the differences within the N-terminal region. Whereas the average per-residue turn propensities are very similar for A$\beta_{1-38}$ and A$\beta_{1-43}$, among the four isoforms, A$\beta_{1-40}$ ensembles exhibit the highest turn propensity in the R5-D7 region, and A$\beta_{1-42}$ ensembles are characterized by the highest turn propensities in the A2-F4 region. The average per-residue strand propensities exhibit several isoform-specific characteristics (Figure 3B). Overall, these propensities are the highest in the A$\beta_{1-38}$ conformational ensemble across the entire sequence. The A2-F4 region is characterized by high strand propensities in A$\beta_{1-38}$, A$\beta_{1-40}$, and A$\beta_{1-43}$ but not in A$\beta_{1-42}$ conformations, where this region is practically strand-free. Unlike the conformations of the other three isoforms, A$\beta_{1-40}$ conformations show very low strand propensities in the R5-E12 region. A$\beta_{1-38}$ conformations have higher strand propensities within C-terminal regions G29-G33 and M35-G37 than the conformations of the other isoforms. The average per-residue coil propensities exhibit, as expected, high values primarily at the N-terminal D1 and the C-terminal G38, V40, A42, and T43 in A$\beta_{1-38}$, A$\beta_{1-40}$, A$\beta_{1-42}$, and A$\beta_{1-43}$ conformations, respectively (Figure 3C). There are two isoform-specific regions with coil propensities exceeding 0.5, i.e., in the A2-F4 region of A$\beta_{1-42}$ and the G9-E11 region of A$\beta_{1-40}$ conformations (Figure 3C). Apart from the differences in the coil propensities at the C-terminus, which are mostly due to the sequence differences among the four isoforms, conformations of A$\beta_{1-42}$ are uniquely characterized by predominant per-residue coil propensity in the A2-F4 region (exceeding 0.7), where the conformations of the other three isoforms show considerable strand per-residue propensities in this region (Figure 3B,C).

We then asked whether the average turn, strand, and coil content per peptide depend on the assembly state. To this end, the ensembles of conformations are sorted out by the assembly state, and the average turn, strand, and coil content per peptide are calculated as described in Section 2. Figure 4 shows the average turn, strand, and coil content versus the oligomer order for all four Aβ isoforms.

A$\beta_{1-38}$ and A$\beta_{1-42}$ monomers have comparable average turn content, which is significantly lower than the turn content in A$\beta_{1-40}$ and A$\beta_{1-43}$ monomers (Figure 4A). For each of the four isoforms, there is a significant increase in the turn content upon dimer formation; however, all oligomers up to heptamers are characterized by comparable average turn contents. Overall, the average turn content is the highest in oligomers formed by A$\beta_{1-40}$, followed by A$\beta_{1-42}$, A$\beta_{1-43}$, and A$\beta_{1-38}$, which has the lowest turn content. The latter is easy to understand because A$\beta_{1-38}$ lacks the C-terminal turn region, which is present in the other three isoforms (Figure 3A). Figure 4B demonstrates that for all four isoforms, the strand content is the highest in monomers, whereby A$\beta_{1-38}$ and A$\beta_{1-42}$ monomers have the highest strand content (∼0.50), followed by A$\beta_{1-40}$ monomers with a significantly lower strand content (∼0.38) and A$\beta_{1-43}$ monomers with the least strand content (∼0.27). For all four isoforms, the strand content decreases upon dimer formation and does not depend considerably on the oligomer order. Overall, as shown in Figure 4B, oligomers formed by A$\beta_{1-38}$ are characterized by the highest strand (∼0.30), followed by oligomers formed by A$\beta_{1-43}$ (∼0.26), A$\beta_{1-42}$ (slightly above ∼0.20), and A$\beta_{1-40}$ (slightly below ∼0.20). The coil content is the lowest in monomers, ranking from the lowest to the highest from A$\beta_{1-38}$, A$\beta_{1-42}$, A$\beta_{1-40}$, to A$\beta_{1-43}$, respectively, (Figure 4C). The largest increase in the coil content occurs upon A$\beta_{1-38}$, A$\beta_{1-40}$, and A$\beta_{1-42}$ dimer formation, whereas the coil content is independent of the assembly state in A$\beta_{1-43}$ conformations. A$\beta_{1-42}$ oligomers are characterized by a significantly higher coil content than the oligomers formed by the other three isoforms (Figure 4C), followed by A$\beta_{1-40}$, A$\beta_{1-43}$, and A$\beta_{1-38}$ oligomers with the least coil content. The increased coil content in oligomers formed by A$\beta_{1-42}$ (relative to those formed by the other three isoforms) is consistent with the results of the average per-residue coil propensity shown in Figure 3 and indicates that this high coil content originates mostly from the high per-residue coil propensities in the region A2-F4. If we use the average coil content to quantify the degree of intrinsic disorder in Aβ oligomers, then A$\beta_{1-42}$ oligomers display the most intrinsic disorder, indicating the highest level of structural plasticity among the four isoforms under study.

### 3.4. Isoform-Specific Tertiary and Quaternary Structures of Aβ Oligomers


We next analyzed the tertiary and quaternary structures of oligomers up to hexamers formed by the four Aβ isoforms. For each assembly state, we derived two types of contact maps, intramolecular and intermolecular contact maps (see Section 2 for details), displaying tertiary and quaternary contacts, respectively (Appendix A and Appendix A). Notably, the probabilities of formation of specific tertiary (Appendix A) or quaternary (Appendix A) contacts are ∼0.1 and do not exceed 0.3, consistent with an intrinsically disordered nature of Aβ isoforms forming oligomers without well-defined (specific) contacts. To elucidate the peptide regions involved in tertiary contact formation, Appendix A elucidates tertiary contacts that the A2-F4 region forms with the other peptide regions by a magenta rectangle, whereas tertiary contacts between the CHC and MHR or CTR are enclosed in a gray rectangle and tertiary contacts that the CTR makes with the MHR are enclosed in a black triangle. Similarly, in Appendix A, the quaternary contacts between pairs of CHCs are enclosed in gray rectangles, quaternary contacts between the CHC and MHR or CTR can be found within magenta rectangles, and the quaternary contacts involving the CTR and MHR are enclosed in black triangles. Overall, Appendix A and Appendix A show that the CHC, MHR, and CTR of Aβ are significantly more involved in tertiary and quaternary contact formation than other regions, as expected, due to a predominance of hydrophobic residues in these regions. Moreover, contact maps of a specific Aβ isoform exhibit self-similarity, indicating that tertiary and quaternary structures are not strongly affected by the oligomer order. Intra- and intermolecular contact maps show some isoform specificity. Notably, the relative prominence of the CHC in self-assembly (Appendix A, gray rectangles) is the strongest for A$\beta_{1-38}$ oligomers and decreases with the length of the peptide. Of all four isoforms, A$\beta_{1-42}$ is characterized by the strongest involvement of the C-terminal region in oligomer formation (Appendix A, black triangles).

To facilitate a more detailed comparison of tertiary and quaternary contacts among the four isoforms, we selected dimers and hexamers as representative oligomers and calculated pairwise differences of the respective intra- and intermolecular contact maps: A$\beta_{1-38}$–A$\beta_{1-40}$, A$\beta_{1-42}$–A$\beta_{1-40}$, A$\beta_{1-43}$–A$\beta_{1-40}$, and A$\beta_{1-43}$–A$\beta_{1-42}$, as shown in Figure 5.

The isoform specificity of intramolecular contact maps is elucidated in difference maps displayed in Figure 5, columns 1 and 3, for dimers and hexamers, respectively. Notably, the observed nonzero contacts are mostly due to the primary structure differences at the C-terminus. Nonetheless, there are a few noteworthy observations that reflect the effect of the C-terminal sequence on the tertiary structure. Some tertiary contacts in A$\beta_{1-38}$ dimers are stronger than in the respective A$\beta_{1-40}$ dimers, F4-Y10, H13-V18, V24-A30, and G29-G37, whereas tertiary contacts L18-V40, F19-V40, I31-V40, and V36-V40 that are strong in A$\beta_{1-40}$ dimers do not exist in A$\beta_{1-38}$ dimers. Similarly, some tertiary contacts in A$\beta_{1-38}$ hexamers are stronger than in the respective A$\beta_{1-40}$ hexamers (F4-Y10, L17-V36, K28-G37, G29-G38, and I31-V36), whereas the three strongest tertiary contacts, I31-V40, M35-V39, and V36-V39, in A$\beta_{1-40}$ hexamers have no counterparts in A$\beta_{1-38}$ hexamers. This comparison reveals that A$\beta_{1-38}$ oligomers compensate for the lack of tertiary contacts at the C-terminus by recruiting the more N-terminal regions relative to A$\beta_{1-40}$ oligomers. Such a compensation of the shorter variant is less obvious when tertiary contacts in A$\beta_{1-42}$ dimers and hexamers are compared to those in A$\beta_{1-40}$ dimers and hexamers. Aβ42 dimers and hexamers are dominated by the C-terminal residues (V40, I41, and A42), forming strong tertiary contacts with other hydrophobic residues along the sequence (F4, V18, F19, I31, I32, G33, and M35, V36), whereas these contacts do not exist in A$\beta_{1-40}$ dimers and hexamers. A$\beta_{1-40}$ oligomers form a larger number of somewhat stronger tertiary contacts than those formed by A$\beta_{1-42}$ oligomers within the more N-terminal regions, such as A2-L34 and V18-I31 in A$\beta_{1-40}$ dimers and F20-V24 in A$\beta_{1-40}$ hexamers. Very similar observations can be made with respect to the tertiary structure comparison between A$\beta_{1-43}$ and A$\beta_{1-40}$ oligomers, where A$\beta_{1-43}$ oligomers are characterized by strong tertiary contacts involving the C-terminal V40, I41, and A42 (V18-I41, F19-I41, I31-I41, I32-A42, M35-V40), which do not exist or do not form in A$\beta_{1-40}$ oligomers. In contrast, tertiary contacts that are somewhat stronger in A$\beta_{1-40}$ than in A$\beta_{1-43}$ oligomers are L17-V39, and V18-V39, V18-I31 in dimers and F20-V24, A21-V24, and G25-G29 in hexamers, respectively. A tertiary structure comparison between A$\beta_{1-43}$ and A$\beta_{1-42}$ oligomers reveals less apparent differences than those described above. Nonetheless, A$\beta_{1-43}$ oligomers exhibit a few stronger tertiary contacts than A$\beta_{1-42}$ oligomers involving the N-terminus, such as E3-S8, for example, while tertiary contact I32-I41 is significantly stronger in A$\beta_{1-42}$ than in A$\beta_{1-43}$ dimers. Overall, the above results show that the prominence of the C-terminal region of Aβ increases with the length of the peptide, i.e., the C-terminal region plays an increasingly important role in the stabilization of the tertiary structure as the peptide length increases.

The isoform specificity of intermolecular contact maps is elucidated in difference maps displayed in Figure 5, columns 2 and 4, for dimers and hexamers, respectively. Intermolecular contacts between pairs of CHC regions, which have been reported to stabilize $A\beta_{1-40}$ oligomers [23], are even stronger in $A\beta_{1-38}$ dimers and hexamers (Figure 5, row 1, columns 2 and 4). $A\beta_{1-38}$ hexamers are characterized by a line of stronger diagonal intermolecular contacts than $A\beta_{1-40}$ hexamers, indicating an increased tendency for a parallel alignment of peptides within a hexamer (Figure 5, row 1, column 4). The $A\beta_{1-42}$–A$\beta_{1-40}$ difference maps of intermolecular contacts reveal a similar pattern, with A$\beta_{1-40}$ displaying a higher tendency to adopt a parallel alignment than A$\beta_{1-42}$ both in dimers and even more so in hexamers, where this tendency is spread along the entire Aβ sequence (Figure 5, row 2, column 4). The $A\beta_{1-43}$–A$\beta_{1-40}$ difference map of intermolecular contacts follow a similar pattern as the $A\beta_{1-42}$–A$\beta_{1-40}$ difference map (Figure 5, row 3, columns 2 and 4). A comparison of quaternary contacts in $A\beta_{1-42}$ and $A\beta_{1-43}$ reveals stronger contacts in $A\beta_{1-43}$ than in $A\beta_{1-42}$ dimers (Figure 5, row 4, column 2), whereas $A\beta_{1-42}$ hexamers exhibit stronger contacts than $A\beta_{1-43}$ hexamers (Figure 5, row 4, column 4). The analysis of the quaternary structure above reveals isoform-specific characteristics of intermolecular contacts, indicating that shorter isoforms exhibit a greater propensity for a parallel alignment of peptides within the oligomers, and this tendency is more pronounced in hexamers than in dimers.

### 3.5. Free Energy Landscapes Reveal Isoform Specificity of Conformational Ensembles


We here investigate the oligomer-size-specific free energy landscapes by evaluating the potential mean force (PMF) for all four variants, A$\beta_{1-38}$, A$\beta_{1-40}$, A$\beta_{1-42}$, and A$\beta_{1-43}$, from dimers to hexamers by employing the N-C distance and hydrophobic SASA as the two reaction coordinates. The hydrophobic SASA quantifies the exposure of hydrophobic residues to the solvent, which is expected to be related to the self-assembly propensity, whereas the N-C distance is a measure of how extended each peptide is within the oligomer. We employ two different PMF calculations with respect to the N-C distance. Appendix A is based on the N-C distance for each individual peptide within an oligomer to elucidate the variability of this quantity within oligomers as well as among oligomers. On the other hand, Figure 6 is based on an average value of the N-C distance within an oligomer of a specific size, resulting in smoother free energy landscapes, which we also used to identify the lowest free energy conformations, displayed in Figure 6. These conformations range in morphology from quasi-spherical for dimer, trimer, and tetramer conformations to more elongated and ellipsoidal for pentamer and hexamer conformations, akin to morphologies observed by atomic force microscopy imaging of A$\beta_{1-40}$ and A$\beta_{1-42}$ self-assembly during the lag phase before the onset of amyloid fibril formation [52]. Notably, as reported previously for A$\beta_{1-40}$ pentamer and hexamer conformations [29], we observe dumbbell-shaped morphologies among the characteristic A$\beta_{1-38}$ conformations (Figure 6).

Figure 6 reveals two tendencies of the free energy landscapes that are in common to all four Aβ isoforms. First, the hydrophobic SASA decreases with the oligomer size. This change is the most significant between dimer and trimer free energy landscapes. This is not surprising because in larger oligomers, the hydrophobic regions can be more efficiently concealed from implicit water, reducing their surface area exposed to the solvent. Second, the average N-C distance within an oligomer tends to increase with oligomer size, indicating that individual peptides tend to become more extended in larger oligomers. Among the four isoforms, A$\beta_{1-42}$ stands out as it is characterized with the most variable reaction coordinates within dimers through hexamers, indicating the largest extent of structural flexibility (Figure 6). These same features are present in the free energy landscapes in Appendix A, where all per-peptide N-C distances within oligomers are included.

The free energy landscapes in Figure 6 and Appendix A show a high degree of isoform specificity. Of the four isoforms, A$\beta_{1-40}$ forms the most compact conformations with the most localized minima, whereas A$\beta_{1-42}$ exhibits the shallowest and most dispersed minima, followed by A$\beta_{1-43}$ and A$\beta_{1-38}$. Interestingly, A$\beta_{1-38}$ is characterized by significantly less compact oligomers than A$\beta_{1-40}$ with, on average, more extended peptides within oligomers. Similarly, A$\beta_{1-43}$ trimers through hexamers exhibit less variable reaction coordinates than the respective A$\beta_{1-42}$ oligomers. These observations indicate that the above free energy landscapes are not sensitive exclusively to the peptide length but display complex sequence specificity that can be difficult to predict without employing computer simulations.

### 3.6. Isoform-Specific Characteristics of Oligomer Conformations


We here explore the distance between the N-terminus (NT) or C-terminus (CT) of each peptide within an oligomer conformation and the center of mass (CM) of the oligomer. The two respective distributions of the NT-CM and CT-CM distances offer valuable insights into the elongation and overall shape of the resulting conformations. The probability distributions of the NT-CM and CT-CM distances for monomers through hexamers formed by A$\beta_{1-38}$, A$\beta_{1-40}$, A$\beta_{1-42}$, and A$\beta_{1-43}$ in Figure 7 indicate significant differences among the four isoforms as well as dependence on the oligomer size.

The NT-CM distance distributions in Figure 7 (left column) for all four peptides show that the average NT-CM distance increases with the oligomer size, indicating that the N-termini are more effectively excluded from the CM of larger oligomers. The NT-CM distance distributions for monomers show the most isoform specificity, whereby the average NT-CM distance is the smallest for A$\beta_{1-38}$, followed by A$\beta_{1-43}$, A$\beta_{1-40}$, and A$\beta_{1-42}$. The NT-CM distance distributions for A$\beta_{1-38}$, A$\beta_{1-40}$, and A$\beta_{1-43}$ dimers almost overlap, whereas the A$\beta_{1-43}$ distributions of the NT-CM distance for trimers and larger oligomers are on average shifted to larger distances than the respective A$\beta_{1-40}$ and A$\beta_{1-38}$ distributions. Overall, the two shorter isoforms, A$\beta_{1-38}$ and A$\beta_{1-40}$, exhibit on average the shortest NT-CM distances for all conformations, monomers through hexamers, indicating that their N-termini are in closer proximity to the CM of the conformations than in conformations formed by the two longer isoforms, A$\beta_{1-42}$ and A$\beta_{1-43}$. Among the four isoforms, A$\beta_{1-42}$ stands out because its NT-CM distance distributions extend to the largest NT-CM distances for all conformations, consistent with solvent-exposed and flexible N-termini that characterize A$\beta_{1-42}$ conformations.

The CT-CM distance distributions in Figure 7 (right column) show an even larger extent of isoform specificity than the respective NT-CM distance distributions for all conformations, from monomers through hexamers. The CT-CM distance distributions of oligomers formed by A$\beta_{1-38}$ are extended to larger distances than the respective distributions of the other three isoforms. The C-terminus of A$\beta_{1-38}$ lacks four hydrophobic residues spanning residues 39-42 (VVIA), which are present in the other three peptides. This primary structure characteristic of A$\beta_{1-38}$ may be the reason why the C-terminal residue is farther away from the CM of the hydrophobically collapsed oligomer conformations. Moreover, we noted that A$\beta_{1-38}$ pentamers and hexamers tend to form dumbbell conformations (Figure 6), which also contributes to increased CT-CM distances. The CT-CM distances in A$\beta_{1-43}$ oligomers are, on average, larger than the respective distances in A$\beta_{1-40}$ and A$\beta_{1-42}$ oligomers. This may be due to threonine at 43, which is not a hydrophobic residue and may thus interfere with the tendency of the VVIA region to form a compact oligomer core. On average, the shortest CT-CM distances in dimers through pentamers are associated with A$\beta_{1-40}$ conformations, followed by A$\beta_{1-42}$ conformations. Interestingly, across hexamers formed by the four isoforms, A$\beta_{1-42}$ is associated with lowest CT-CM distances, even lower than A$\beta_{1-40}$, which is likely due to the tendency of A$\beta_{1-40}$ hexamers to adopt dumbbell conformations. This observation elucidates A$\beta_{1-42}$ hexamers as conformations with the most C-terminally stabilized oligomer core.

### 3.7. Fractal-like Morphological Characteristics of Aβ Oligomers


While Figure 6 shows characteristic morphologies of dimers through hexamers of the four peptides under study, the oligomer size distributions in Figure 2 and Appendix A indicate that larger oligomers formed as well, albeit with lower propensities. Figure 8 shows examples of the largest assemblies formed by the four isoforms: a 14-mer of A$\beta_{1-38}$, 15-mer of A$\beta_{1-40}$, 22-mer of A$\beta_{1-42}$, and 18-mer of A$\beta_{1-43}$. These assemblies resemble Aβ assemblies with curvilinear “beads-on-a-string” morphologies, often observed by atomic force microscopy [52].

To examine whether or not there are any significant morphological differences among oligomers formed by different variants, the three principal moments of inertia, $I_{1}>I_{2}>I_{3}$, were derived for monomers through decamers of each of the four peptides, following the procedure reported previously [31]. The results in Appendix A (panel A) indicate that all three principal moments of inertia scale with the oligomer size approximately as a power law. Figure 6 shows that oligomers up to hexamers across all four peptides adopt similar quasi-spherical to ellipsoidal morphology, which explains why $I_{2}\approx I_{1}$, whereas $I_{3}$ is smaller than $I_{1}$ and $I_{2}$ (Appendix A, panel B). Monomers, dimers, and trimers exhibit mostly quasi-spherical shapes as indicated by rather high $I_{3}/I_{1}$ and $I_{3}/I_{2}$ ratios. These two ratios steadily decrease by a factor of two between tetramers and heptamers and remain on average unchanged for octamers through decamers. These observations suggest that tetramers through heptamers undergo an abrupt elongation. Notably, A$\beta_{1-42}$ pentamers through heptamers, nonamers, and decamers are characterized by significantly larger $I_{3}/I_{1}$ and $I_{3}/I_{2}$ ratios than the respective oligomers formed by the other three variants. This can be explained by considering that the N-termini of A$\beta_{1-42}$ oligomers are the most flexible and extend, on average, further away from the long axis of the oligomer than the N-termini of oligomers formed by the other three isoforms.

A more quantitative analysis can be conducted by assuming an ellipsoidal shape of a typical oligomer of length *L* and thickness *D*. The elongation and thickening scaling exponents, α and β, can be defined as $L∼n^{\alpha }$ and $D∼n^{\beta }$, where *n* is the number of Aβ peptides in the assembly. We can then extract these two exponents from the scaling properties of $I_{i}∼n^{\alpha_{i}}$ (i=1,2,3) by considering that $I_{1}\approx I_{2}∼ML^{2}$ and $I_{3}∼MD^{2}$, where *M* is the mass of the assembly and $M=nM_{0}$ (and $M_{0}$ is the mass of a monomer). Appendix A (panel A) shows the log–log plots of $I_{i}$ and the respective linear regression analysis for oligomer sizes that exhibit clear power-law behavior, i.e., tetramers through decamers for $I_{1}$ and $I_{2}$ and monomers through hexamers for $I_{3}$. The corresponding exponents, $\alpha_{i}$, are reported in Appendix A alongside the isoform-specific values of the resulting elongation exponent $\alpha =(\alpha_{1}+\alpha_{2})/4-1/2$, the thickness exponent $\beta =\alpha_{3}/2-1/2$, and the scaling of the volume of assemblies, $V∼LD^{2}∼n^{\gamma }$, where $\gamma =\left(\alpha_{1}+\alpha_{2}\right)/4+\alpha_{3}-3/2$. Note that the arithmetic mean of $\alpha_{1}$ and $\alpha_{2}$ is used in the above derivation of the scaling exponent equations. The elongation scaling exponent α in Appendix A is more than two-fold larger than the thickness scaling exponent β, demonstrating the tendency of oligomers to elongate rather than increase in thickness, at least for tetramers, pentamers, and hexamers where the two power-law regimes overlap. Similarly, these oligomers are fractal-like as their volume scaling exponent γ>1. Interestingly, all the scaling exponents exhibit isoform specificity, with the lowest value of γ associated with A$\beta_{1-42}$, suggesting that of the four isoforms, A$\beta_{1-42}$ forms the most compact tetramers, pentamers, and hexamers.

## 4. Conclusions

Here we examine oligomer formation of A$\beta_{1-38}$, A$\beta_{1-40}$, A$\beta_{1-42}$, and A$\beta_{1-43}$, using DMD4B-HYDRA simulations, which were previously applied to the two predominant alloforms, A$\beta_{1-40}$ and A$\beta_{1-42}$ [23], their N-terminally truncated variants [30], Arctic mutants [29,45], and [K16A] and [K28A] mutants [31]. A new protocol, utilizing 32 independent trajectories per system, is applied to improve on the statistical analysis aiming to elucidate the effect of four different Aβ variants that differ at the C-terminus, including A$\beta_{1-38}$ and A$\beta_{1-43}$, which were not studied by this method before, on self-assembly and the resulting conformational ensembles; while all four isoforms are present in the human brain, two of the four isoforms under study have been associated with neuroprotective effects (A$\beta_{1-38}$ and A$\beta_{1-40}$) and the other two with increased toxicity (A$\beta_{1-42}$ and A$\beta_{1-43}$).

Starting from 32 unstructured and spatially separated monomers for each of the 32 replicas per isoform, we show that the oligomer size distributions evolve into quasi-stable distributions after 20 $\times \;10^{6}$ simulation time units (about 0.6 μs). The conformations extracted from the simulations between 20 $\times \;10^{6}$ and 40 $\times \;10^{6}$ simulation time units (corresponding to 0.6–1.2 μs) are used for structural analysis. As expected, this new protocol reproduces A$\beta_{1-40}$ and A$\beta_{1-42}$ oligomer size distributions that are consistent with the previously reported distributions [23,29] and experimental findings [32,54]. The comparison of oligomer size distributions among the four variants reveals that the oligomer size distributions of the two shorter peptides, A$\beta_{1-38}$ and A$\beta_{1-40}$, are dominated by smaller oligomers: dimers, trimers, and tetramers (in the case of A$\beta_{1-40}$). The two longer peptides, A$\beta_{1-42}$ and A$\beta_{1-43}$, exhibit, on average, increased propensities to form hexamers and larger oligomers, such as decamers through pentadecamers. Voelker et al. examined fully atomistic A$\beta_{1-40}$ and A$\beta_{1-42}$ monomers through pentamers in water, reporting that trimers and larger oligomers form water-permeable pores with a propensity that sharply increases with the oligomer size [51]. If embedded into a cellular membrane, such pore-rich oligomers could act as ion channels, causing calcium influx and subsequent cell death [55]. The results reported by Voelker et al. combined with the findings of this study thus imply that the ability to form larger oligomers is associated with an increased propensity for pore formation. Thus, A$\beta_{1-42}$ and A$\beta_{1-43}$, which form larger oligomers than the two shorter peptides, are expected to exhibit a higher propensity for aberrant ion channel formation, which is likely linked to toxic function. These findings suggest that a potential therapeutic strategy that would break oligomers into smaller assemblies may be able to reduce the pore-forming capacity and thus toxicity mediated by Aβ oligomers.

The secondary structure elements that dominate oligomers formed by all four peptides are turns and β-strands, without any significant helical contributions. All conformations exhibit significant coil content, i.e., the absence of any secondary structure, consistent with the intrinsically disordered nature of Aβ peptides. A$\beta_{1-38}$ conformations have the most β-strand content, followed by A$\beta_{1-43}$ conformations, whereas A$\beta_{1-42}$ conformational ensembles exhibit the highest coil content, rendering A$\beta_{1-42}$ the most intrinsically disordered. The dominant features of the tertiary and quaternary structures of oligomer conformations, such as the involvement of hydrophobic peptide regions in contact formation, are shared among the four peptides, as expected. Yet, tertiary and quaternary structures show remarkable isoform specificity. First, the C-terminal region plays an increasingly important role in the stabilization of the tertiary structure as the peptide length increases. A$\beta_{1-38}$ and to a lesser extent A$\beta_{1-40}$ oligomers compensate for the lack of tertiary contacts at the C-terminus by recruiting the hydrophobic residues at the CHC and at the N-terminus. Second, the role of the CHC region in oligomer formation is the most important in A$\beta_{1-38}$ and diminishes as the length of the peptide increases. Third, the C-terminal region plays the most prominent role in A$\beta_{1-42}$ oligomer formation, although A$\beta_{1-43}$ has an additional residue at the C-terminus T43. This is most likely because unlike the C-terminal sequence VVIA of A$\beta_{1-42}$, T is not a hydrophobic residue, which seems to interfere with oligomer stabilization.

The analysis of the free energy landscapes provides insights into the shapes of oligomers, which range from quasi-spherical and ellipsoidal to curvilinear “beads-on-a-string” conformations. For example, A$\beta_{1-38}$ has a high propensity for formation of dumbbell-like pentamer and hexamer morphologies, akin to those previously reported for A$\beta_{1-40}$ hexamers [29]. The NT-CM and CT-CM distance distributions demonstrate that oligomer morphologies are strongly isoform-specific. A$\beta_{1-42}$ stands out among the four isoforms because the NT-CM distance distributions extend to the largest values for all oligomer sizes under study, consistent with solvent-exposed and flexible N-termini that characterize specifically A$\beta_{1-42}$ oligomers. Furthermore, the CT-CM distance distributions are both isoform- and oligomer-size specific, whereby A$\beta_{1-40}$ monomers through pentamers exhibit, on average, the smallest distances, followed by A$\beta_{1-42}$ conformations. Interestingly, among hexamers of all four isoforms, A$\beta_{1-42}$ hexamers show on average the lowest CT-CM distances, indicating that these oligomers are optimally stabilized through quaternary contacts among the C-termini. These conclusions on the isoform-specific morphology of oligomers are further supported by quantitative analysis of the three principal moments of inertia, which allowed us to derive the elongation and thickness scaling exponents, revealing the fractal nature of oligomer structures formed by the four Aβ isoforms. Inspection of the largest oligomers formed by the four peptides in our simulations also indicates the polymorphic nature of larger oligomers, showing curved as well as branched oligomers that may be interesting to explore in the future.

The structural origin of Aβ oligomer-mediated toxicity is still unknown. Solvent-exposed and flexible N-termini of Aβ oligomers have been previously associated with increased toxicity in the context of AD [56]. Of the four Aβ isoforms studied here, only A$\beta_{1-42}$ oligomers carry this feature, which offers a plausible explanation for increased toxicity of A$\beta_{1-42}$ relative to A$\beta_{1-40}$ oligomers [57,58,59]. Previously, DMD4B-HYDRA simulations were used to examine oligomer formation of two Arctic mutants, [E22G]A$\beta_{1-40}$ and [E22G]A$\beta_{1-42}$, linked to the early-onset familial AD (FAD) [29]. This FAD mutation resulted in [E22G]A$\beta_{1-40}$ and [E22G]A$\beta_{1-42}$ oligomers with disordered and solvent-exposed N-termini akin to those found in A$\beta_{1-42}$ oligomers, which could help explain increased toxicity mediated by Arctic mutants in this early-onset FAD [29]. In addition, oligomer formation by four N-terminally truncated Aβ isoforms (A$\beta_{3-40}$, A$\beta_{3-42}$, A$\beta_{11-40}$, A$\beta_{11-42}$), associated with naturally occurring pyroglutamated Aβ isoforms, was examined by DMD4B-HYDRA simulations to reveal that (i) all four N-terminally truncated peptides formed, on average, larger oligomers and (ii) A$\beta_{3-40}$ and A$\beta_{3-42}$ (but not A$\beta_{11-40}$ and A$\beta_{11-42}$) oligomers featured disordered and solvent-exposed N-termini [30]. These observations provided a plausible explanation for increased toxicity of pyroglutamated A$\beta_{3-40}$ and A$\beta_{3-42}$ relative to the wild-type peptides [30]. These findings combined offer substantial albeit circumstantial evidence that disorder and solvent exposure of N-termini in Aβ oligomers contribute to increased toxicity in AD. However, structural plasticity due to disordered and solvent-exposed N-termini alone may not be sufficient to mediate Aβ oligomer toxicity. This present work shows that A$\beta_{1-43}$ forms oligomers that are on average larger than those formed by A$\beta_{1-42}$, which is expected to increase the pore formation propensity of the former [51]. Thus, both the ability to form large oligomers and N-terminal flexibility and solvent exposure may contribute to the ability of Aβ oligomers to disrupt the cellular membrane and cause cell dysfunction. Therapeutic agents that simultaneously reduce oligomer size and bind to the N-termini of Aβ oligomers may thus represent the best strategy against toxicity mediated by Aβ oligomers.

## Figures and Tables

**Figure 1 biomolecules-14-00774-f001:**
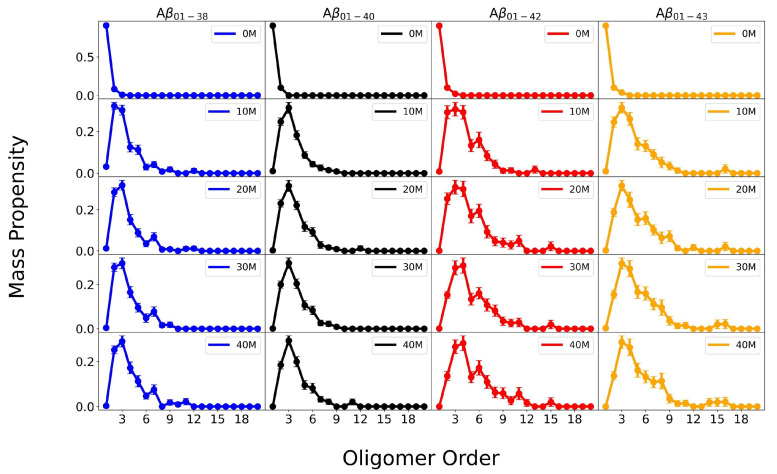
Time evolution of mass-weighted oligomer size distributions of A$\beta_{1-38}$, A$\beta_{1-40}$, A$\beta_{1-42}$, and A$\beta_{1-43}$ using DMD4B-HYDRA simulations. Each oligomer size distribution is an average over size distributions derived for each of 32 independent trajectories. The error bars correspond to SEM values.

**Figure 2 biomolecules-14-00774-f002:**
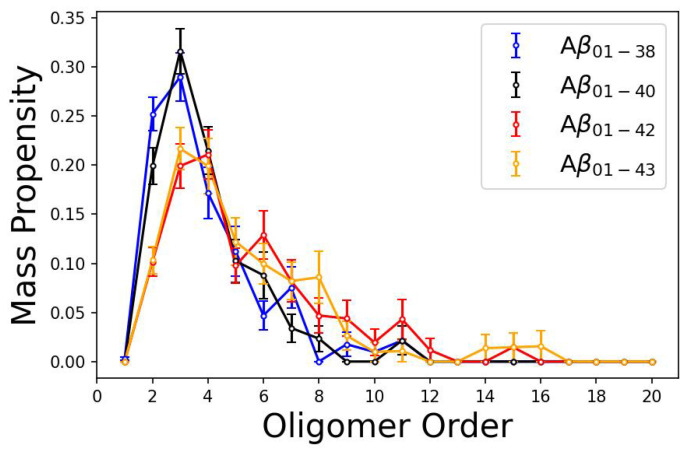
Mass-weighted oligomer size distributions of A$\beta_{1-38}$, A$\beta_{1-40}$, A$\beta_{1-42}$, and A$\beta_{1-43}$ at $40\times 10^{6}$ simulation time units. The error bars correspond to SEM values as described in Section 2.

**Figure 3 biomolecules-14-00774-f003:**
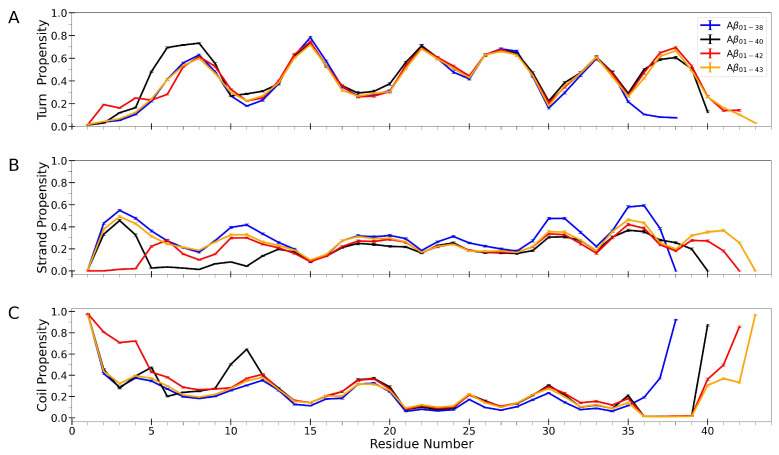
Average per-residue turn, strand, and coil propensities in A$\beta_{1-38}$, A$\beta_{1-40}$, A$\beta_{1-42}$, and A$\beta_{1-43}$ conformational ensembles. The error bars correspond to SEM values.

**Figure 4 biomolecules-14-00774-f004:**
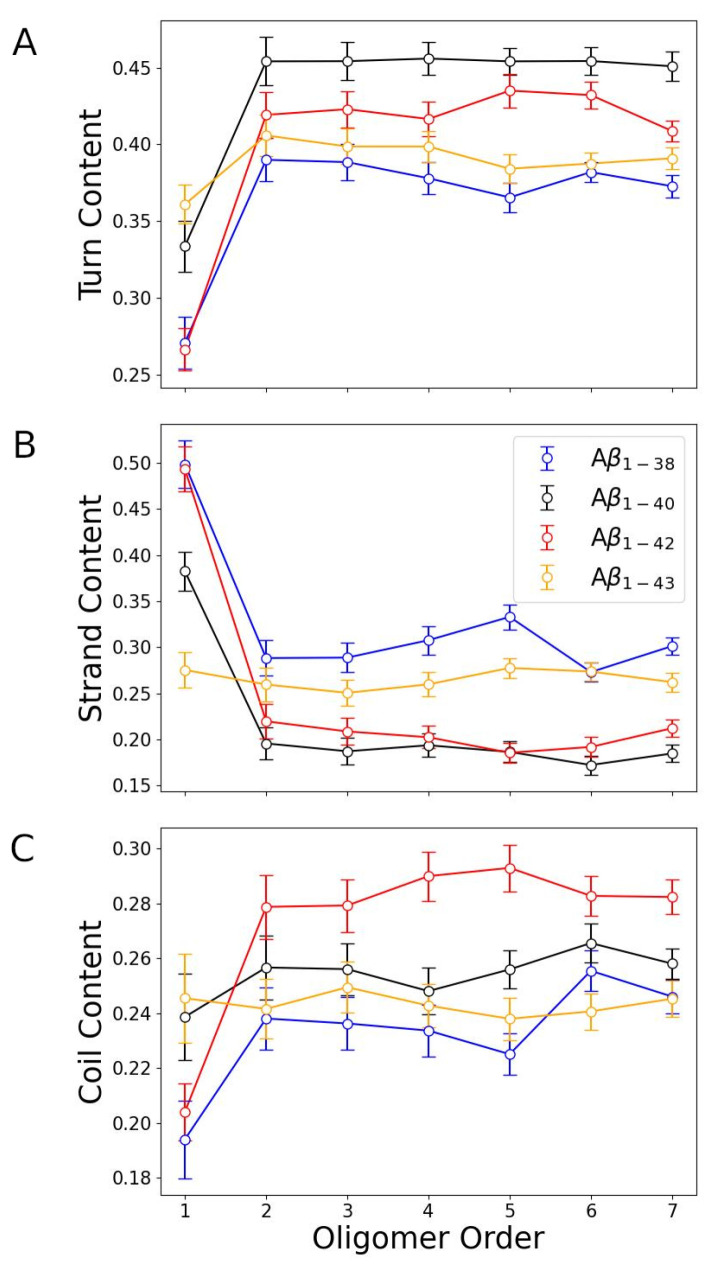
The average turn, strand, and coil content in A$\beta_{1-38}$, A$\beta_{1-40}$, A$\beta_{1-42}$, and A$\beta_{1-43}$ conformations versus the oligomer order. The error bars correspond to SEM values.

**Figure 5 biomolecules-14-00774-f005:**
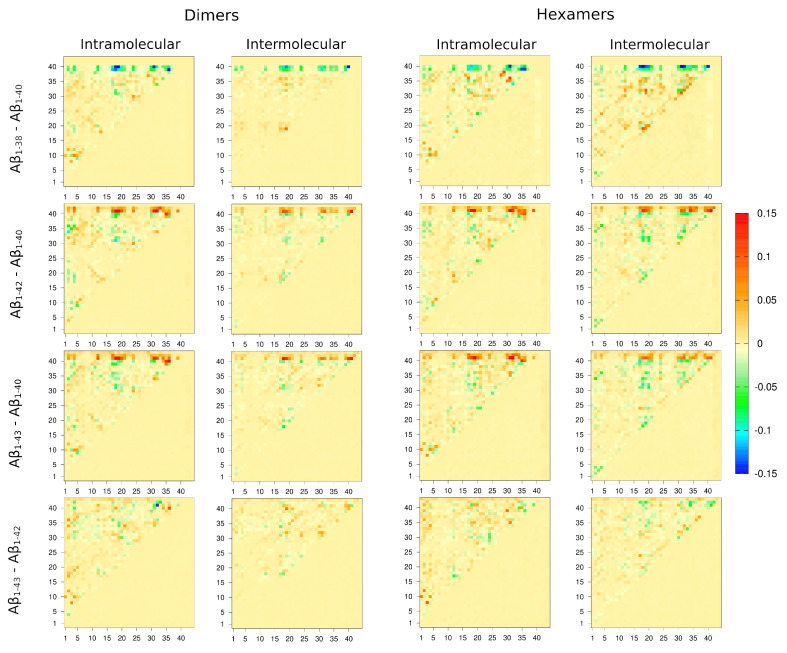
Pairwise contact map differences for dimers and hexamers of A$\beta_{1-38}$, A$\beta_{1-42}$, and A$\beta_{1-43}$ relative to A$\beta_{1-40}$, and A$\beta_{1-43}$ relative to A$\beta_{1-42}$.

**Figure 6 biomolecules-14-00774-f006:**
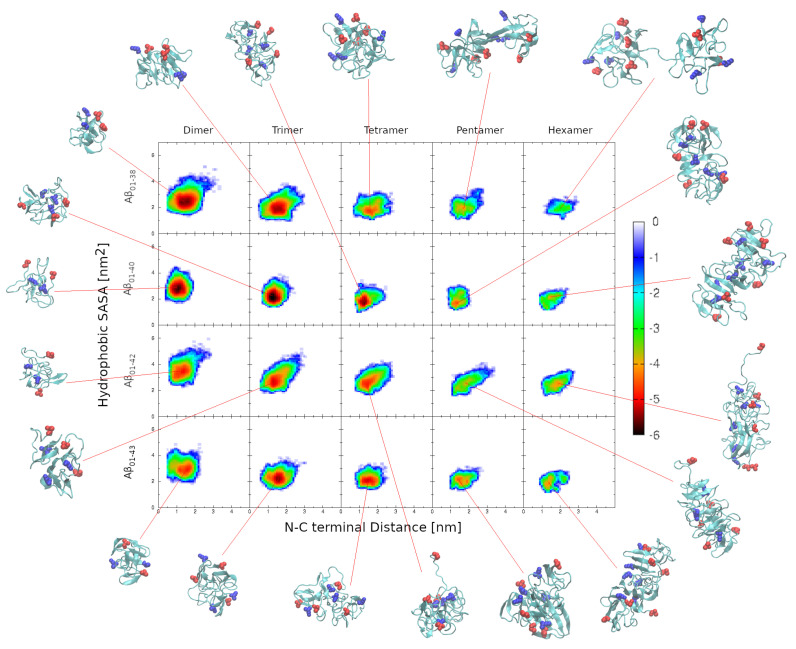
Free energy landscapes of A$\beta_{1-38}$, A$\beta_{1-40}$, A$\beta_{1-42}$, and A$\beta_{1-43}$ dimers through hexamers. The N-C distance and hydrophobic SASA are used as the two reaction coordinates. The color scale is expressed in units of $k_{B}$T.

**Figure 7 biomolecules-14-00774-f007:**
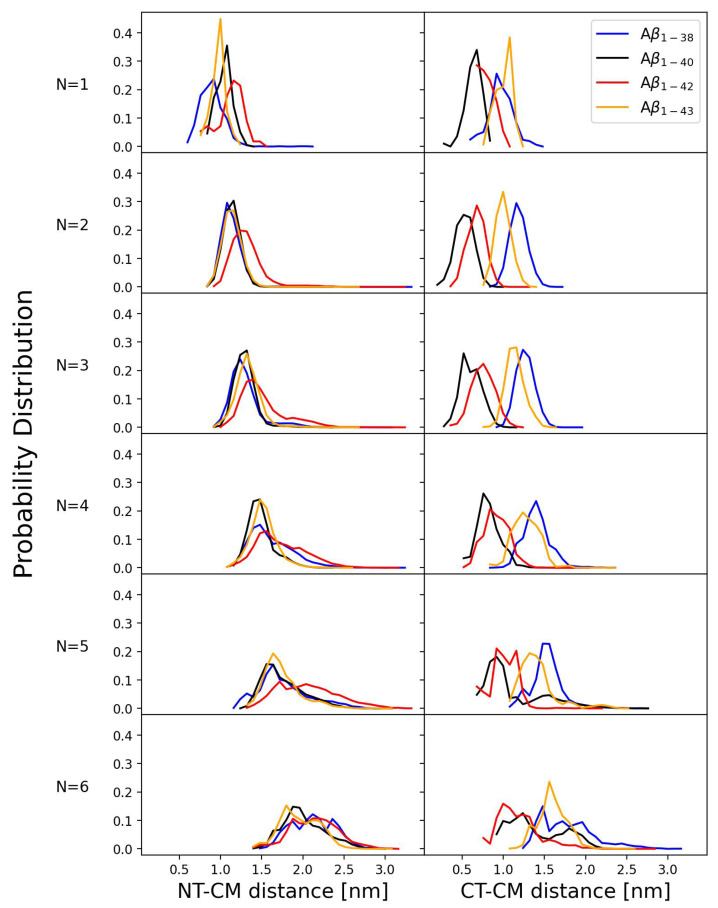
The probability distributions of the distance from C terminal and N terminal to the center of mass for monomers through hexamers of all four variants of amyloid beta peptides: A$\beta_{1-38}$, A$\beta_{1-40}$, A$\beta_{1-42}$, and A$\beta_{1-43}$.

**Figure 8 biomolecules-14-00774-f008:**
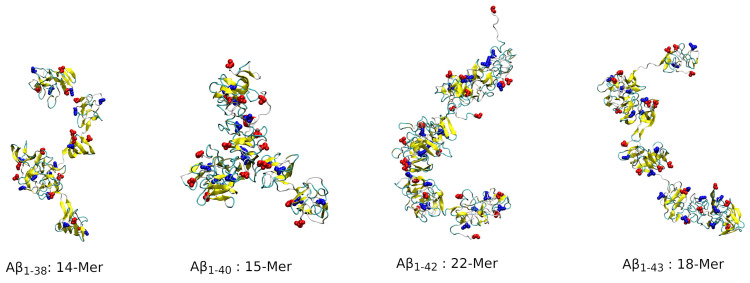
Example of morphologies of the largest oligomers formed by A$\beta_{1-38}$, A$\beta_{1-40}$, A$\beta_{1-42}$, and A$\beta_{1-43}$ in our simulations.

## Data Availability

The data produced in this study are available on request from the corresponding author.

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
