# Peer review of "Oligomer Formation by Physiologically Relevant C-Terminal Isoforms of Amyloid β-Protein"

_biomolecules, 2024, doi:10.3390/biom14070774_

Round 1

Reviewer 1 Report

Comments and Suggestions for Authors

The manuscript authored by Pandey and Urbanc, entitled "Oligomer Formation by Physiologically-Relevant C-Terminal Isoforms of Amyloid β-Protein," investigates the self-assembly of various C-terminally truncated amyloid β-protein (Aβ) isoforms. It focuses on their oligomerization properties and the structural characteristics of these oligomers using discrete molecular dynamics (DMD) simulations. The DMD simulations allowed the authors to analyze oligomer formation in detail, providing valuable insights into the structural dynamics of amyloid β-protein isoforms. The study links structural characteristics of oligomers with pathological features of Alzheimer's disease, enhancing understanding of disease mechanisms. Overall, the manuscript presents a significant advance in the understanding of amyloid β-protein oligomerization, with implications for Alzheimer’s disease research. I recommend acceptance after minor revisions that clarify methodological details and enhance the discussion of clinical implications:

  1. As the authors discussed, the DMD method seems a very powerful tool for studying protein self-assembly. It would be beneficial if the authors could provide more background on DMD methods and related models, along with general applications in the broader field of protein dynamics, misfolding, and aggregation.

  2. The discussion should be expanded to relate the specific findings of this study to broader therapeutic strategies, possibly suggesting how these findings could guide the development of inhibitors targeting specific oligomer forms.

Author Response

The reviewer's comments and our point by point responses are provided in the attached pdf file.

Reviewer 2 Report

Comments and Suggestions for Authors

The authors present the result of discrete simulations of self-assembly for Aβ1−38 , Aβ1−40 , Aβ1−42 and Aβ1−43. It is definitely an interesting paper, where sometimes not entirely expected results are shown. It seems to be scientifically adequate and well carried out.

It is a rather technical article, which sometimes is difficult to read and completely understand the figures presented in order to evaluate whether what the authors say is supported or not.

In general, the text describes the observed results, but further interpretation is lacking. Does the results make senses? How do they correlate with previous results?

Specifically, there are things that could be improved.

- The aim of Table 1 seems to be: To show the number of structures in each conformation that were used in the analyses. Something else?. This could be in the supplementary.

- How valid is the comparison of the different states for each peptide with such a difference in the number of structures used? Aβ1-42 just 56 whereas Aβ1-38 more than 1000.

- If I understand correctly, figure 3 shows the average and error for all conformations in the trajectories. Correct? That the error is so small (I don't see it) implies that the secondary structure does not change practically for any conformation. This seems contradictory with figure 4. In fact, figure 4 has larger errors (Differece for each oligomeric conformation). Does this make sense?

Interestingly, there are secondary structure propensity differences for each peptide globally in Figure 3, but also in Figure 4 we can see that there are differences just due to the oligomeric conformation.

It is possible that the differences that we see in F3 are just due to the fact that there are different conformations for each oligomeric state?.

It would be interesting to seen the data in Figure 3 but separated for each oligomeric state (1 figure for monomer, 1 figure for simmer, etc).

The authors said: “If we use the average coil content to quantify the degree of intrinsic disorder in

Aβ oligomers, then Aβ1− 42 oligomers display the most intrinsic disorder” , however, again if I see figure 3c, I do not observe that, I see that the average per coil residue is low and almost identical between peptides.

- Contact maps are useful for analyzing a large number of data, however, unless you are very familiar with the residues it is difficult to see what interactions are important. It would be useful to draw the most relevant interactions in a diagram of the sequence. See REF17 Fig 3c,d.

- The data for the “two most relevant pests in AD have already been studied with this methodology”, the here it was carried out with greater sampling. However a comparison with the previous data is missing. Did you see anything different? Or is everything the same regardless of the greater sampling?

- What about other pepties already studied. The new data help to understand or complement in some way the data reported for the peptides with the N-terminally truncated (REF 19)?

- The conclusions, more than conclusions, are a summarized repetition of what was already said in the paper.

Author Response

The reviewer's comment and our point by point responses are provided in the attached pdf file.
